# Word Importance Explains How Prompts Affect Language Model Outputs

## Abstract

The emergence of large language models (LLMs) has revolutionized numerous applications across industries. However, their "black box" nature often hinders the understanding of how they make specific decisions, raising concerns about their transparency, reliability, and ethical use. This study presents a method to improve the explainability of LLMs by varying individual words in prompts to uncover their statistical impact on the model outputs. This approach, inspired by permutation importance for tabular data, masks each word in the system prompt and evaluates its effect on the outputs based on the available text scores aggregated over multiple user inputs. Unlike classical attention, word importance measures the impact of prompt words on arbitrarily-defined text scores, which enables decomposing the importance of words into the specific measures of interest–including bias, reading level, verbosity, etc. This procedure also enables measuring impact when attention weights are not available. To test the fidelity of this approach, we explore the effect of adding different suffixes to multiple different system prompts and comparing subsequent generations with different large language models. Results show that word importance scores are closely related to the expected suffix importances for multiple scoring functions. We plan to make the Python project for computing these scores available on GitHub and discuss how it could assist developing generative Artificial Intelligence (AI) use-cases.

**Keywords**: Large Language Models, Explainability, Masking, Word Importance.

## 1 Introduction

Large language models (LLMs) have become the focal point of contemporary computational linguistics and artificial intelligence research. With their capacity to generate human-like text, comprehend complex linguistic patterns, and perform tasks across multiple domains, LLMs have shown tremendous potential in applications ranging from chatbots to content generation. However, with their increased capabilities come challenges—principally, the challenge of explainability. The opaque nature and large parameter space of these models poses a profound concern[1][2], not only for researchers attempting to understand and optimize them but also for end-users who seek transparency and reliability in their outputs, see Glikson and Woolley (2020).

Explainability in machine learning (ML), or the ability to understand and interpret model decisions, has become a topic of utmost importance, see Došilović et al. (2018), Xu et al. (2019), Gohel et al. (2021) and Danilevsky et al. (2020b). As decisions made by these models influence an ever-expanding range of sectors, from healthcare to finance, the need for model interpretability has grown exponentially. Understanding how an LLM arrives at a particular output is not just a matter of scientific interest but has broader societal implications, especially when considering issues of bias, fairness, and accountability, see Li et al. (2023), Gallegos et al. (2023) and Bender et al. (2021).

---

[1]https://openai.com/research/language-models-can-explain-neurons-in-language-models
[2]https://openaipublic.blob.core.windows.net/neuron-explainer/paper/index.html

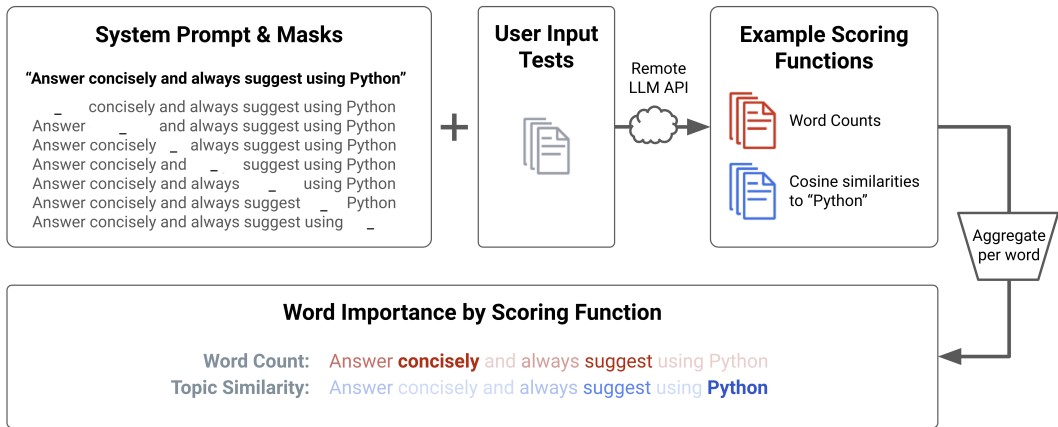

Figure 1: Illustration of the word importance method. Words from the system prompt are masked with underscore. The masked system prompts, together with user inputs, are passed to an LLM and the outputs are evaluated with arbitrary text scores. The importance score for every word from the prompt with regard to the selected text score is computed by comparing these results with the results from using the original system prompt.

One avenue that has shown promise in shedding light on the LLM decision-making processes is the study of the effects of individual words or phrases of the input on model output, see Wallace et al. (2021). For example, a practitioner may want to measure if a particular word influences the output in a specific way. Recognizing the impact of words or linguistic structures on LLM outputs can offer a granular understanding of model behavior, providing valuable insights into how information is processed and weighted by the model.

Attention is also commonly used to understand the impact of input sequences (Vaswani et al., 2017), however it has limitations and its use for explainability is contested, see Jain and Wallace (2019) and Serrano and Smith (2019). Attention does not readily indicate in what way the input tokens influence the output. For example, a particular word may have high attention, but not influence the reading-level, social bias, or desired topics of interest in the output. Additionally, attention weights are not available for many of the most popular closed-source models, necessitating alternative means of model analysis[3].

This paper delves into the explainability of LLMs, focusing on the role that individual words play in influencing different characteristics of the model outputs. The experimental setup measures how well word importance statistics correspond to the user intent of different system prompts. We show that this analysis can illuminate the behavior of LLMs, promoting a future where these models are not only effective but also transparent and accountable.

## 2 Related Works

Word importance has been a significant topic of interest in natural language processing (NLP) and has gained even more attention with the rise of large language models (LLMs) Sun et al. (2021). Understanding which words are important in a given context can provide insights into how models make decisions, help improve interpretability, and gain users' trust. There have been multiple works in the area of explainability in NLP, see Danilevsky et al. (2021), Danilevsky et al. (2020a), and Wiegreffe and Pinter (2019).

Wallace et al. (2021) showed that short sequences of trigger words can significantly impact the output of a language model, such as changing the sentiment of an analyzed text from positive to negative or manipulating the model to answer in a harmful way. This shows how important it is to understand the impact of individual words and phrases.

---

[3]Anthropic API: https://docs.anthropic.com/claude/reference/getting-started-with-the-api

Permutation importance is a technique to interpret machine learning models by assessing the impact of individual features on the model's predictive performance (Breiman, 2001). This method involves perturbing the values of a single feature and measuring the subsequent degradation in model performance, thus inferring the importance of that feature.

Another important widely used method for model explainability is SHAP, see Shapley et al. (1953) and Sundararajan and Najmi (2020). SHAP (SHapley Additive exPlanations) is a method for explaining the output of machine learning models. This technique allocates the contribution of each feature to every single prediction, ensuring that the sum of the contributions equals the difference between the prediction and a predefined baseline value. Central to SHAP is the concept of Shapley values, derived from game theory, which ensures a fair distribution of contributions among features by considering all possible combinations of features and the marginal contributions they provide (Lundberg and Lee, 2017). One of the primary advantages of SHAP is its consistency and accuracy in attributing feature importances across a range of ML applications. However, as with all interpretability techniques, it is important to apply SHAP values judiciously, understanding the underlying assumptions and potential nuances in different modeling contexts.

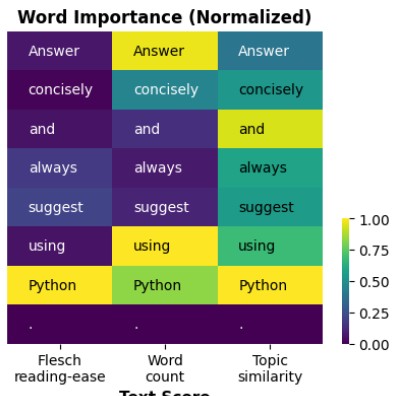

Figure 2: System prompt word importance evaluated by multiple scores. Each word is masked with _ to compute its word importance score. Using multiple scores simultaneously allows one to conveniently observe the multifaceted impact of each word on the model output.

A different area of investigation has focused on efforts to understand the representations that LLMs learn given a query text (Rogers et al. (2020) and Paganelli et al. (2022)). This includes looking at the activations (Hermans and Schrauwen (2013) and Karpathy et al. (2015)), attention weights (Clark et al. (2019), Kovaleva et al. (2019) and Htut et al. (2019)), mutual information (Hoover et al., 2021), probing classifiers (Belinkov (2021), Tenney et al. (2019) and Liu et al. (2019)), shuffling and truncating the text (Ettinger, 2020) and – most similar to this work – masking or perturbing particular words (Wu et al. (2020) and Kim et al. (2019)). These approaches lack the wide applicability of our approach, relying on direct access to the model and particular linguistic assumptions or tasks. In addition, our approach focuses on the higher-level intent of the user contained in the prompt or additional general criteria as opposed to understanding the generation of a particular token.

Finally, Yin et al. (2023) studied the role that different parts of a system prompt have in the context of instruction learning. They find, for instance, that model performance is particularly sensitive to descriptions and examples of the desired model output and suggest that this allows for prompt compression by leaving out other parts. Compared to our word-by-word approach, this study focuses more on the semantics of impactful prompt components.

## 3 WORD IMPORTANCE METHOD

Understanding the significance of individual words within a prompt is crucial for insights into the mechanics of language models and its response as well as for optimizing the effectiveness of the prompt itself. The ability to explain the effects of words and phrases on the LLM response enables users of LLM applications to better understand the responses generated by the LLM. It is important to emphasize that this approach aims to assist with explainability of LLM tools. On the other hand, efforts in prompt engineering aim to modify and adjust prompts to improve LLM responses and ensure alignment with use case needs.

To ascertain the importance of each word in a given prompt, we implemented a method inspired by the permutation importance commonly used in tabular data analysis. It helps to quickly identify the importance of each word with regard to several text scores, see Figure 2. The insights from word importance can guide prompt engineering efforts, see Section 5 for further discussion.

### 3.1 Detailed Steps

Given a system prompt $s$ and a set of $M$ user inputs $U$ per system prompt, the proposed word importance approach involves systematically masking one word $k$ at a time and observing the resulting changes in a user-defined NLP scoring function $f$ based on the model's output $m(s, u)$ for $u \in U$.

The method starts by sampling baseline LLM outputs using the unmodified system prompt and one or more example user prompts per system prompt. Next, one word $k$ in the system prompt $s$ is masked by replacing the word with an underscore character, giving $s_k$, and the LLM is sampled again with the modified system prompt, computing $m(s_k, u)$. Finally, one or more user-defined metric scores $f$ are calculated for each LLM output. The relative importance $w(k)$ of each masked word $k$ is given by computing the absolute value of the difference between the masked-prompt score $f(m(s_k, u_j))$ and the baseline score $f(m(s, u_j))$

$$ w(k) = \frac{1}{N\,M} \sum_{i=1}^{N} \sum_{j=1}^{M} |f(m(s, u_j)) - f(m(s_k, u_j))| , \tag{1} $$

where $N$ is the number of completions generated by the model. This methodology is text score agnostic – any scoring function for assessing the output text can be used to ascertain word importance. Figure 1 illustrates the word importance method.

As can be seen in Figure 1, the model first takes the original system prompt "Answer concisely and always suggest using Python", combined with a user input and generates a baseline scoring response utilizing a scoring function. Then, in the first masking iteration, the word "Answer" in the system prompt is masked, and the model uses "_ concisely and always suggest using Python", combined with user inputs to generate new scoring responses. Comparing the scoring responses with the baseline can lead us to better understanding of the importance of the words to the model and its output. This process is repeated for all the words in the system prompt. Figure 2 shows how the results could be presented. The procedure is detailed as follows:

1. Baseline Calculation: For every combination of system prompt $s$ and user input $u$, the model's output $m(s, u)$ is computed multiple $N$-times to establish baseline scores $f(m(s, u))$. These baselines provide the reference against which changes are measured.

2. Word Masking and Output Generation: Each word[4] in the system prompt is sequentially masked, creating a modified version of the system prompt $s_k$. With the word $k$ masked, the model is tasked $N$-times with generating outputs.

3. Scoring and Impact Calculation: For every generated output $m(s_k, u)$ from the masked input, a score $|f(m(s, u)) - f(m(s_k, u))|$ is derived. This score represents the deviation from the baseline, obtained by computing the absolute value of the change in output score relative to the baseline. An average of these deviation scores across the $N$ iterations provides an "impact score" $w(k)$ for the blanked word $k$, reflecting its relative importance, see Equation 1.[5]

---

[4]This includes stopwords but not special characters like punctuation marks. We included stopwords because we wanted to show that our method works when they are included. We can easily exclude stopwords using a program flag and would do so in practice to reduce the method's cost.

[5]We use $N = 3$. To compute word importance scores more quickly, $N$ could be reduced to one output or, for more reliable scores, increased to five outputs per combination.

The schematic illustration below depicts the methodology. As previously noted, any text evaluation score can be used to determine word importance. The setup section emphasizes the scoring functions employed during our experiments.

---

**Algorithm 1** Word importance

---

**Require:** System prompt $s$, user input $u$, number of completions $N$, scoring functions $F$
  **for each** user input $u$ **do**
    generate baseline completion ($N$-times)
    **for each** word $k$ from $s$ **do**
      create masked prompt $s_k$ by masking word $k$ from system prompt $s$ with _
      generate completion using masked word ($N$-times)
      **for each** scoring functions $f \in F$ **do**
        word score is absolute values of the change in completion score ($N$-times)
      **end for**
    **end for**
  **end for**
  average word score is word importance $w(k)$

---

Algorithm 1: Schematic illustration of the world importance algorithm.

## 4 EXPERIMENTATION AND RESULTS

### 4.1 SETUP

#### DATA COLLECTION AND COMPOSITION

We use two types of datasets: artificial data, and questions from the test dataset of SQuAD 2, see Rajpurkar et al. (2018).

The artificial dataset comprises multiple system prompts, a variety of user inputs (questions) for each system prompt, a list of topics associated with the system prompt, and a list of topics associated with the user question. This dataset which includes impersonations and questions is generated using GPT-4. First we collected 112 generally important topics and then we instructed the model to generate three impersonations as system prompts and three questions as user input for each topic. As input to our experiments, we used pairs of system prompts and user inputs with related topics. Using artificial data has the benefit that we can easily generate impersonations and user inputs from a wide range of topics without posing off-topic questions. A typical instance would be the topic "Healthcare", where the impersonation may read as "You are a nurse". See 7.2, in the appendix, for a list of artificial system prompts and 7.1 for further information on the artificial dataset.

The SQuAD 2 dataset contains human-generated questions and not every question is answerable. This is a more realistic and challenging setup that is selected to further evaluate our method and confirm our results using artificial data.

#### TEXT SCORES

In our experiments, we use Flesch reading-ease (Flesch, 1948), word count, and topic similarity (Aletras and Stevenson, 2014) as example scoring functions.

As our focus is on the general applicability of our method, we selected three different but rather simple text scores. Flesch reading-ease is a common score to measure the readability of a text, word count is an example where the score has no fixed upper limit and can be utilized in cases where length of the response is important. The "topic similarity" scoring function is a measure of how well the output relates to specific topics of interest, which has practical uses in many applications. We define it as the cosine similarity between the two embeddings for the generated text and the topic name of interest, for instance, "AI". For topic similarity we make use of text embeddings, using the all-MiniLM-L6-v2 embedding model from the Hugging Face sentence-transformers Python package. In an ablation study,

see Appendix 7.3, we demonstrate the effectiveness of this approach and compare the selected model with other embedding models. We compute Flesch reading-ease and word count with the help of the Python package textstat.

We selected scores that can detect the impact of the selected suffixes, see below, and are sufficiently diverse to see a stronger impact of a suffix for one text score at a time. This is not deemed strictly imperative within the context of our experimental setup, as the paramount concern lies in elucidating the interplay between the suffix impact and the (maximum) word importance from the suffix. Nonetheless, we believe its incorporation enhances the interpretability of our findings.

SUFFIX CONFIGURATION

For each of the system prompts in the artificial dataset, we introduce a suffix to the system prompt to discern its effect on the output, contrasting with the original prompt, see Appendix 7.1 for an example. We want to show that the maximum of the individual word importance scores from the suffix are positively correlated with the impact score of the complete suffix. More on this in Section 4. In order to capture this, we select three suffixes and three text scores designed to quantify the effect:[6]

- Suffix: "Give a detailed answer" with evaluation score: Word count

- Suffix: "Prefer technical terms" with evaluation score: Flesch reading-ease

- Suffix: "Focus on how [COMPANY] could help" with evaluation score: Topic similarity ([COMPANY])

Every suffix is expected to impact its associated evaluation score: the first suffix is expected to result in more verbose outputs, the second to result in lower reading level outputs, and the third to result in outputs that contain a specific topic.

For questions from the SQuAD 2 dataset, we use a fixed system prompt because we do not control the question topic: "Answer truthfully.". To get a more diverse set of results, we altered the suffixes to the system prompt and used:

- Suffix: "Respond in the form of a long story" with evaluation score: Word count

- Suffix: "Explain the answer like I am five" with evaluation score: Flesch reading-ease

- Suffix: "Describe if there are any relationships to AI research" with evaluation score: Topic similarity (AI)

The first suffix is expected to result in more verbose outputs, the second to result in higher reading level outputs, and the third to result in outputs that are similar to the topic "AI". The table below highlights the evaluation metrics, models, and encoders used.

Table 1: Basic experiment configuration

| Parameter/Category | Details |
| --- | --- |
| Evaluation scores | Readability scores: Flesch reading-ease, Word count, Topic similarity |
| Models Employed | GPT-3.5 Turbo (16K, version 0613) Llama2-13B (llama2-13b-chat-hf) |
| Encoders Used | all-MiniLM-L6-v2 |
| Number of user inputs per prompt | 1 |
| Number of outputs per prompt | 3 |
| Temperature setting | 1 |

---

[6]Note that every output is always scored with each text score.

## 4.2 RESULTS

To assess the effectiveness of our method, we analyze the relationship between the impact of a suffix and the maximum word importance of words from the suffix. If adding the suffix has a great impact, then we generally expect to find a word with a high word importance within the suffix. In many cases, when there is a word with a great word importance within a suffix, the suffix has a large impact as well. However, there are cases where there is a word that has a significant positive impact on the output score while another one has a significant negative impact, resulting in a relatively neutral suffix impact. Therefore, we generally expect observing a positive correlation (denoted as $r$) between the suffix impact and the maximum word importance from the suffix. However, we also anticipate encountering more cases where the maximum word importance is larger than the suffix impact.

In the scatterplots shown here, the $x$-axis captures the maximum word importance score for words in the suffix and the $y$-axis is the impact of the suffix as a whole. We draw a linear interpolation line for each suffix and also state the Pearson correlation coefficients in the plots. For larger plots see Appendix 7.4.

Figure 3: Actual suffix importance vs maximum importance from suffix. For each suffix, the word importance has been calculated using scoring functions "word count", "Flesch reading-ease", and "topic similarity". We can clearly see how the output from GPT-3.5 Turbo clusters into a set of outputs where the suffix impact is roughly of the size of the maximum word importance and into a set where the maximum word importance is significantly greater.

We observe positive correlations for all combinations of datasets, LLMs, suffixes and text scores. We also see that the maximum word importance from the suffix is usually greater than the impact of the suffix as a whole, as expected. One exception is the suffix "Respond in the form of a long story" when added to questions from SQuAD 2, given to GPT-3.5 Turbo, and evaluated using the word count score. In the majority of these instances, the suffix impact is greater than the maximum word importance score. This observation, while not undermining the efficacy of our approach, underscores an intriguing and noteworthy special case wherein the entirety of the suffix exerts a greater impact than the maximum word importance derived from the suffix. The central point of relevance remains the positive correlation between these factors. The likely reason is that "long" and "story" both have a significant positive impact on word count and that GPT-3.5 Turbo needs the full suffix for this to take effect. This result suggests further research into multi-word masking approaches will be fruitful.

For Llama2-13B, we have too few results to claim with certainty that the method works as well for Llama2-13B as for GPT-3.5 Turbo but the overall trend is the same and there are no conflicting results.

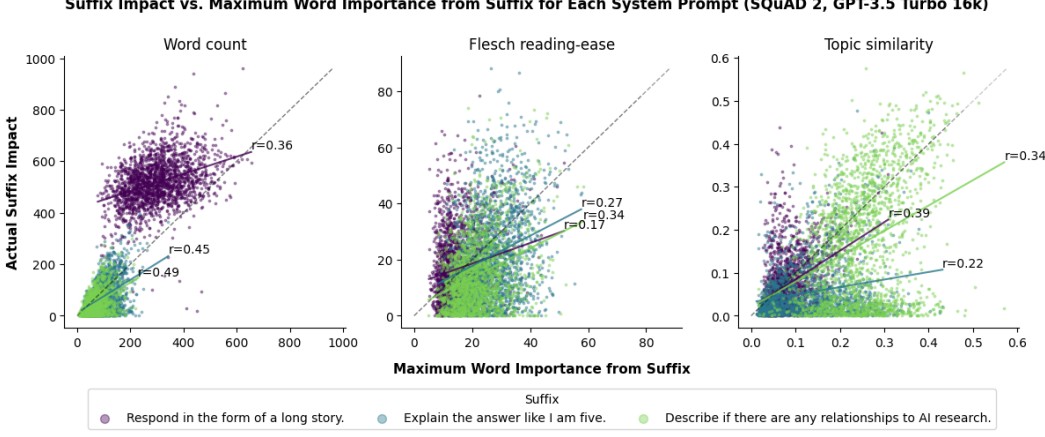

Figure 4: Actual suffix importance vs maximum importance from suffix. For each suffix, the word importance has been calculated using scoring functions "word count", "Flesch reading-ease", and "topic similarity".

Figure 5: Actual suffix importance vs maximum importance from suffix. For each suffix, the word importance has been calculated using scoring functions "word count", "Flesch reading-ease", and "topic similarity".

## 5 LIMITATIONS AND FUTURE DIRECTIONS

An advantage of the "word importance" method is that it is simple to implement, can be applied to proprietary models, and is related to well-understood techniques in the data science domain. Masking individual words gauges the word's value in its contribution to the final output. A higher impact score indicates a stronger influence of that word on the model's output, and vice versa.

However, it's essential to consider factors like the choice of $N$ and the impact of the end user's query on the fully materialized prompt and the system prompt importance scores. Depending on the choice of user input, a model's attention to a word from the suffix will change (Vaswani et al., 2017). Therefore, we have variability of importance scores across runs. Further experiments might include variations in the masking method, such as substituting words rather than merely masking them, to offer a more nuanced understanding of word importance or to optimize for a directional impact with regard to a text score.

Figure 6: Actual suffix importance vs maximum importance from suffix. Each column in this plot represents a suffix mentioned in the table above. For each suffix, the word importance has been calculated using scoring functions "word count", "Flesch reading-ease", and "topic similarity".

In addition, here we only consider insights on how the system prompt affects the outputs for given user prompts and exclude the effect of words in the user prompt on the output or the combination of the two. However, this approach could be extended to analyze the effect of the words of a single materialized prompt. For example, this method could use substitute words for those that are masked by using another model to estimate useful replacement words and their probabilities. Once the cycles of masking and revealing are finished, we could calculate scores for the outputs using both the initial and adjusted prompts, then evaluate the results to gauge the influence of the specific word.

Finally, we could use this method to develop a hierarchical approach to quickly evaluate long prompts and reduce costs: we might start with masking whole sections. Depending on the section importance, we could continue and mask paragraphs, then sentences and finally words.

## 6    CONCLUSION

The "word importance" methodology provides an insight into the internal dynamics of LLMs, particularly regarding how specific words in system prompts influence model outputs. As LLMs permeate various industry applications, understanding their decision-making mechanisms is essential for transparency, accountability, and optimization. A small but diverse set of text scoring functions were evaluated, indicating that this approach may extend to a wide variety of text evaluation metrics, such as human feedback reward models, or estimates of factual accuracy. The methodology, inspired by permutation importance in tabular data, provides a pathway for users and developers to interpret model decisions, thereby paving the way for more ethically designed and understood AI systems. Explainability methods such as this can improve the trust in generative systems in different industry sectors and verticals. For example, in the finance sector, an application to generate financial reports could generate overly optimistic or pessimistic projections in response to certain system prompts. By employing the "word importance" method, stakeholders can identify which words in the prompts significantly influence these outputs in different ways and utilize alternative words in prompts to ensure more neutral and accurate outputs, and inform practitioners on the potential biases introduced by certain prompt words. This would enhance the reliability of the generated reports and foster trust among users by providing transparency into how the model operates.

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

# 7 APPENDIX

## 7.1 ARTIFICIAL DATA

Below is a sample of the experimental data used for assessment. The full dataset is available at: [BLINDED GITHUB URL]

| You answer like Al Gore. | Climate Change | What are the benefits of a plant-based diet? | Nutrition |
|---|---|---|---|
| You answer like Greta Thunberg. | Climate Change | How can we reduce plastic waste? | Environmental Conservation |
| You answer like David Attenborough. | Climate Change | What are the most effective ways to conserve w... | Water Conservation |
| You answer like Bill Nye. | Climate Change | What are the advantages of renewable energy so... | Renewable Energy |
| You answer like Jane Goodall. | Climate Change | How can we protect endangered species? | Wildlife Conservation |

If we are taking the first example, without a suffix, we would provide the following prompt to the model:

- System prompt: You answer like Al Gore.
- User input: What are the benefits of a plant-based diet?

When using the suffix "Give a detailed answer.", this would change to:

- System prompt: You answer like Al Gore. Give a detailed answer.
- User input: What are the benefits of a plant-based diet?

## 7.2 SYSTEM PROMPTS

Below is a sample of system prompts used in this experiment:

- You are a low-income worker sharing your thoughts on economic inequality.

- You are a surgeon.
- You are a wildlife ranger.
- You are an environmental consultant.
- You are a philanthropist working to alleviate global poverty.
- You are a hydroelectric plant operator.
- You are a school principal.
- You are an astronaut.
- You are an environmental scientist.
- You are a teacher.
- You are a geneticist.
- You are a wealthy business owner discussing economic inequality.
- You are a food aid worker.
- You are a feminist activist.

## 7.3 TOPIC SIMILARITY

We define topic similarity of a generated text with a given topic as semantic similarity of the text with the topic name. To compute the topic similarity we use an embedding model to generate embeddings for the generated text and the topic name. Then we compute the cosine similarity between the two embeddings. Here we provide an ablation study to analyze the usefulness of different embeddings models. We selected three embedding models:

- the popular all-MiniLM-L6-v2 from the Hugging Face sentence-transformers library,
- FlagEmbedding (bge-large-en) from the Beijing Academy of Artificial Intelligence (BAAI),[7] and
- the multilingual mBERT (bert-base-multilingual-cased).

We generated an artificial dataset by first selecting the topics "sports", "science", "education", "music", and "phishing". Then we generated seven example paragraphs per topic using GPT-4. To demonstrate that the cosine similarity of the paragraph embedding with the topic embedding can be used as topic similarity, we need to show that the similarity between a topic and its example paragraphs is significantly higher than the similarity between the topic and other examples. Here we use heatmap plots where we align the topics on the $x$-axis and the example paragraphs on the $y$-axis such that we expect relatively high similarity for the diagonal blocks and less for the off-diagonal blocks. Although the top-scoring bge-large-en has overall the highest similarities, the differences between diagonal and off-diagonal blocks are not as significant as for the widely used all-minilm-l6-v2. We would have liked to see a good result using mBERT because it is a multilingual model and could be used when applying this approach to other languages than English, but it showed the worst performance overall and should not be used.

We think that this ablation study shows that our topic similarity score is useful but it would be interesting to see a wider study of the topic, for example, some of the top-scoring embedding models from the MTEB leaderboard might be even better suited.

---

[7]The model was top-ranking on the MTEB leaderboard at the time of the study, see MTEB leaderboard on Hugging Face at https://huggingface.co/spaces/mteb/leaderboard

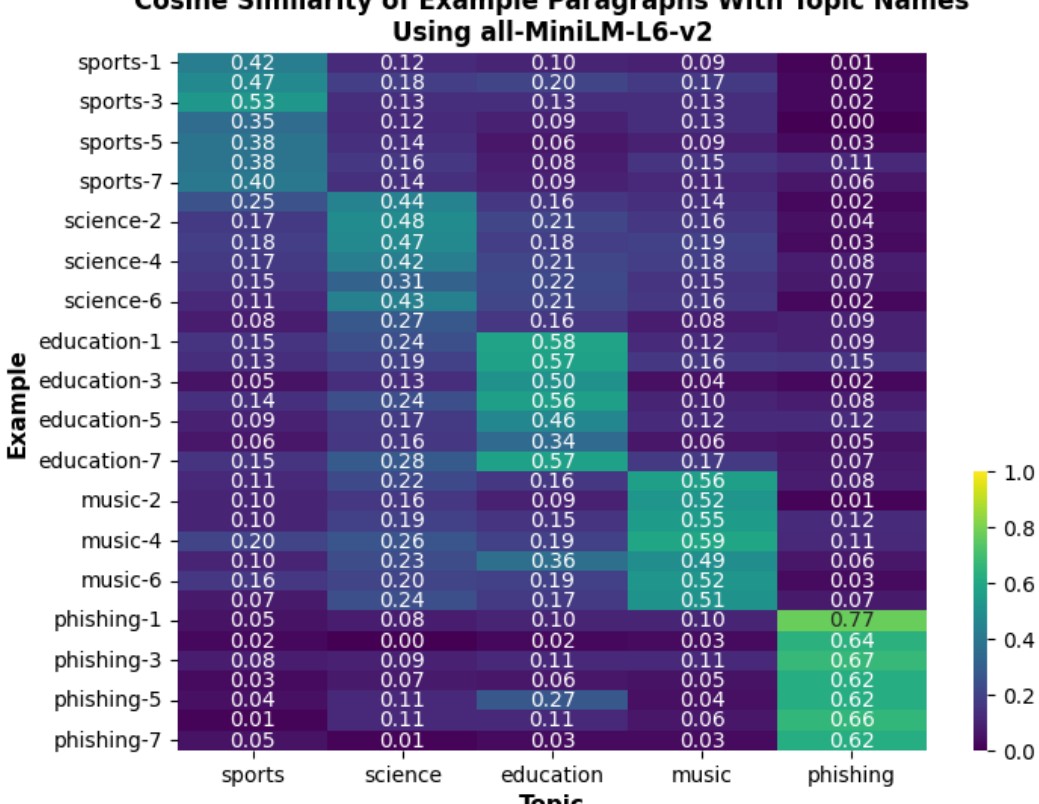

Figure 7: Cosine similarity between topic names and example sentences when using all-minilm-l6-v2 for text embeddings. The difference between diagonal and off-diagonal blocks is most pronounced when using all-minilm-l6-v2 for text embeddings.

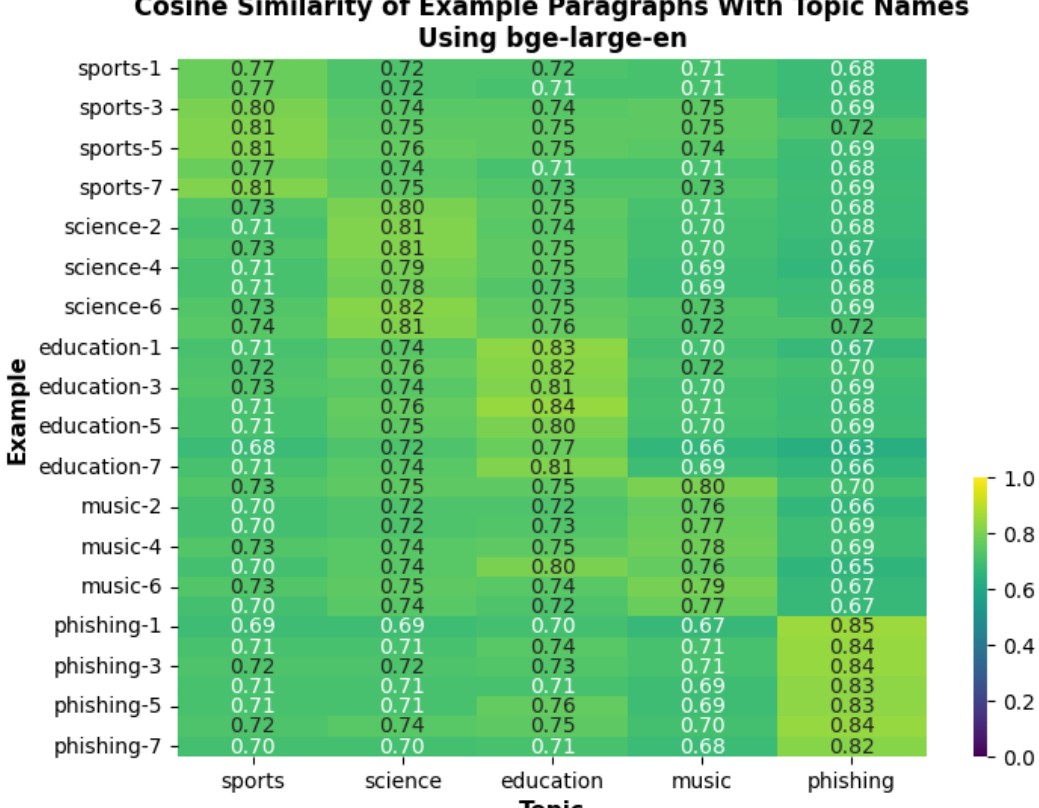

Figure 8: Cosine similarity between topic names and example sentences when using bge-large-en (FlagEmbedding) for text embeddings. We see a high similarity between example sentences and its corresponding topic names (diagonal blocks) when using this embedding model but the difference between diagonal and off-diagonal blocks is not as pronounced as for all-minilm-l6-v2.

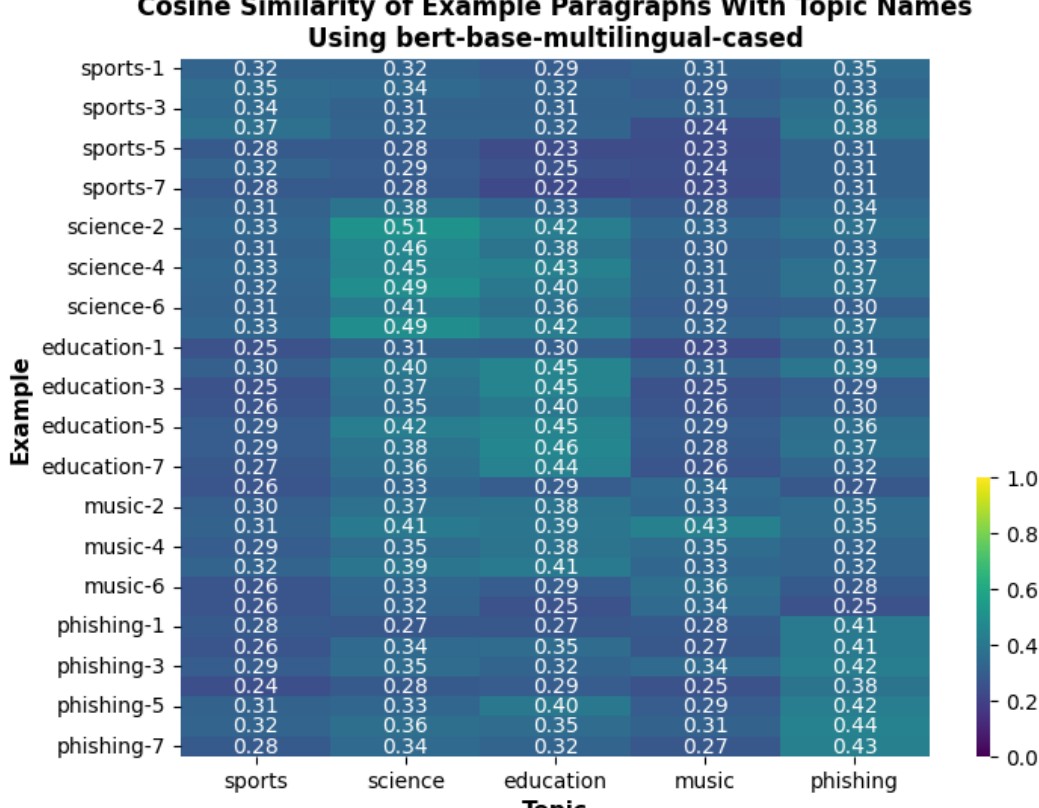

Figure 9: Cosine similarity between topic names and example sentences when using mBERT for text embeddings. The model fails to clearly separate the relevant topic from other topic names.

## 7.4 LARGE RESULT PLOTS

Here are our main results presented again using larger plots. Every row of a large plot corresponds to one subplot from Subsection 4.2.

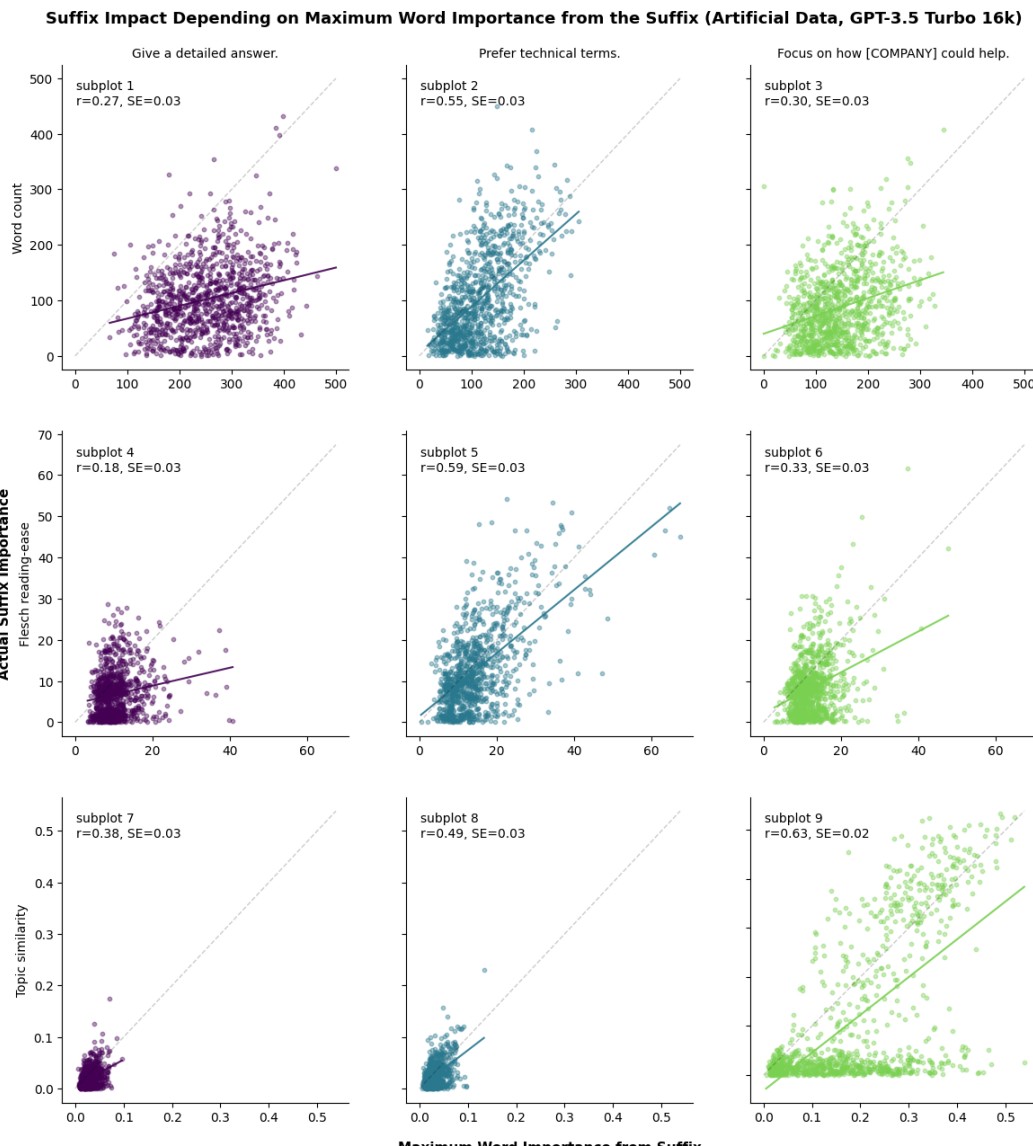

Figure 10: Actual suffix importance vs maximum importance from suffix using GPT-3.5 Turbo on artificial data. Each column in this plot corresponds to one suffix. For each suffix, the word importance has been calculated using scoring functions "word count", "Flesch reading-ease", and "topic similarity".

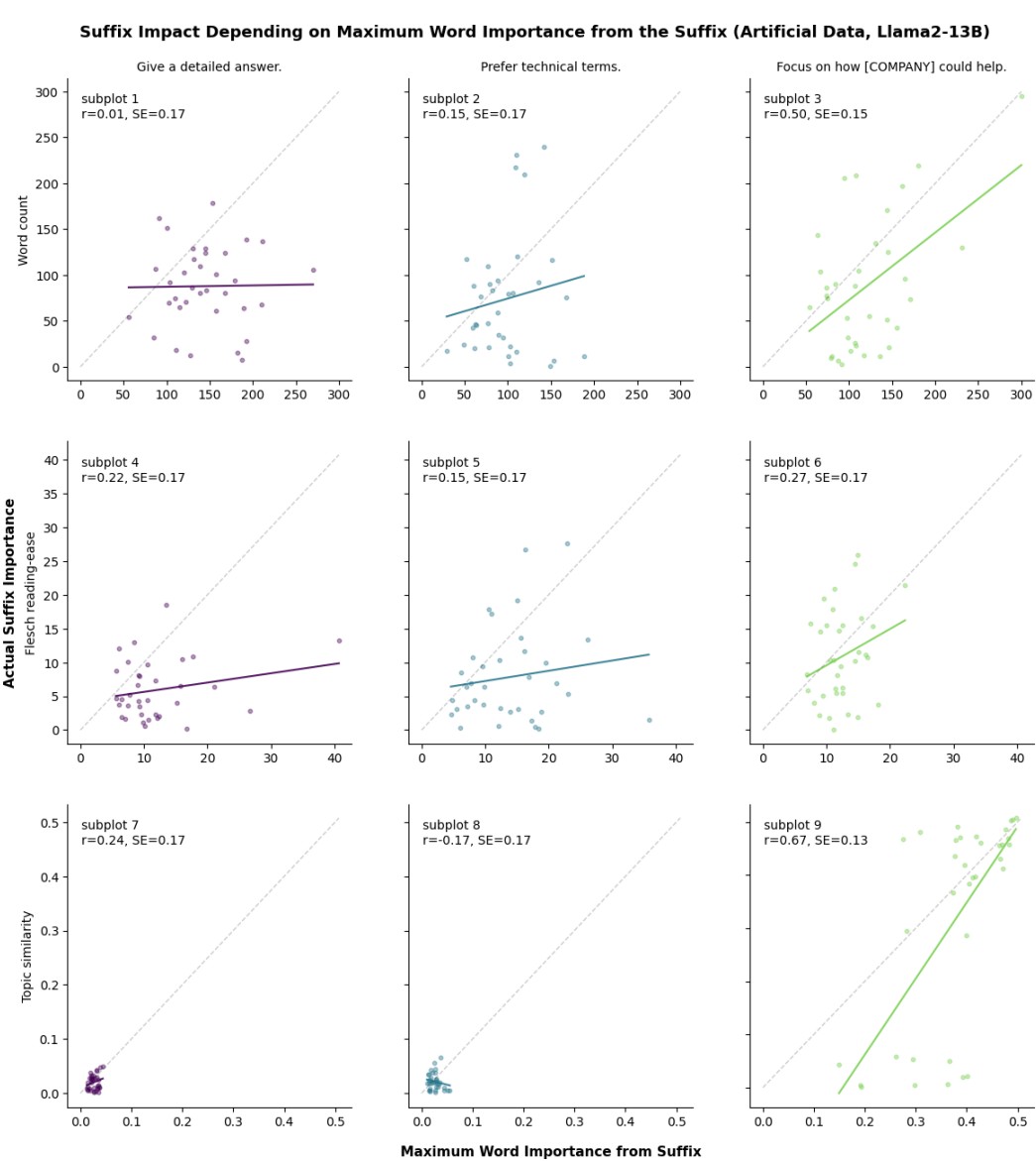

Figure 11: Actual suffix importance vs maximum importance from suffix using Llama2-13B on artificial data. Each column in this plot corresponds to one suffix. For each suffix, the word importance has been calculated using scoring functions "word count", "Flesch reading-ease", and "topic similarity".

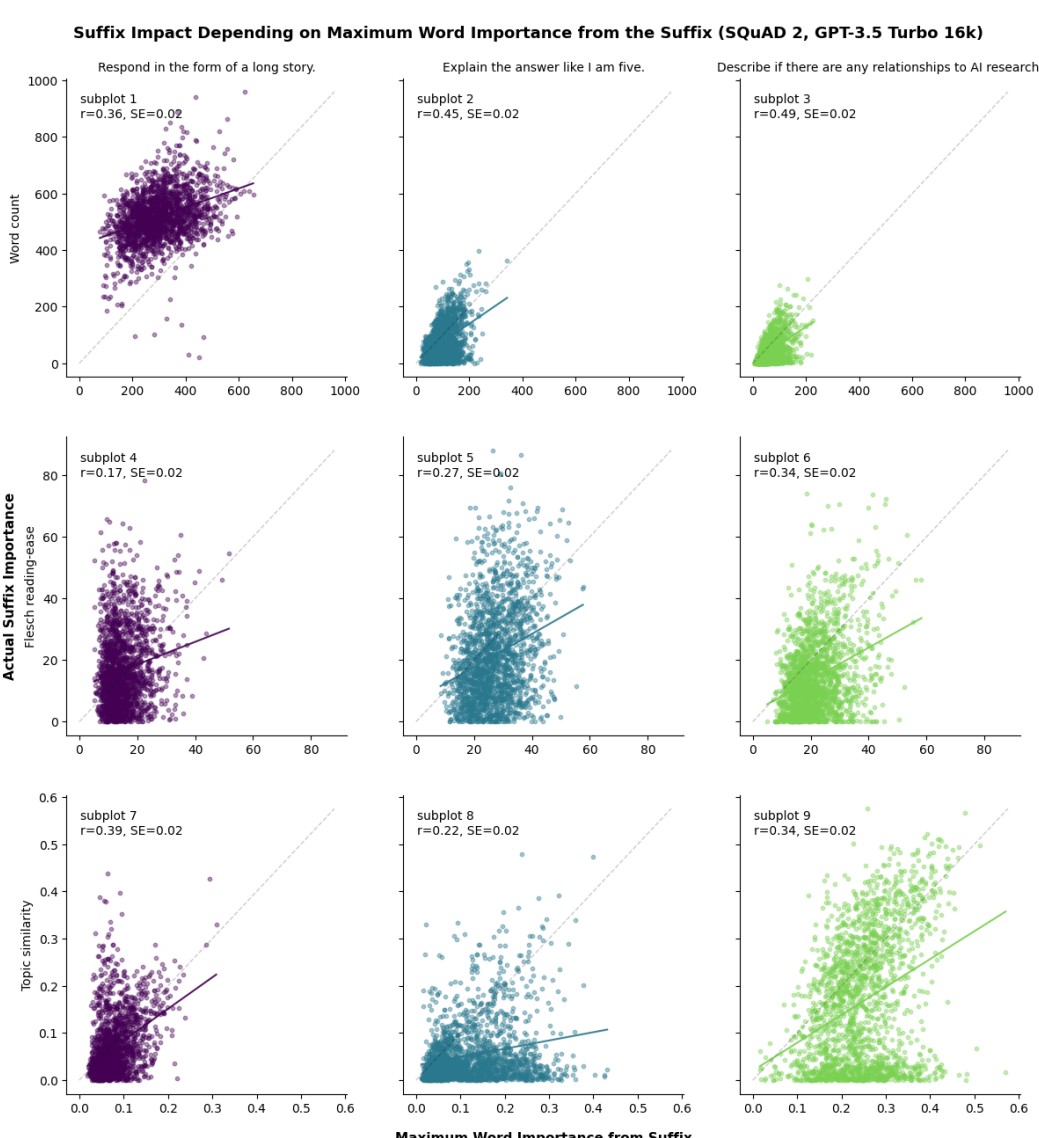

Figure 12: Actual suffix importance vs maximum importance from suffix using GPT-3.5 Turbo on question from SQuAD 2. Each column in this plot corresponds to one suffix. For each suffix, the word importance has been calculated using scoring functions "word count", "Flesch reading-ease", and "topic similarity".

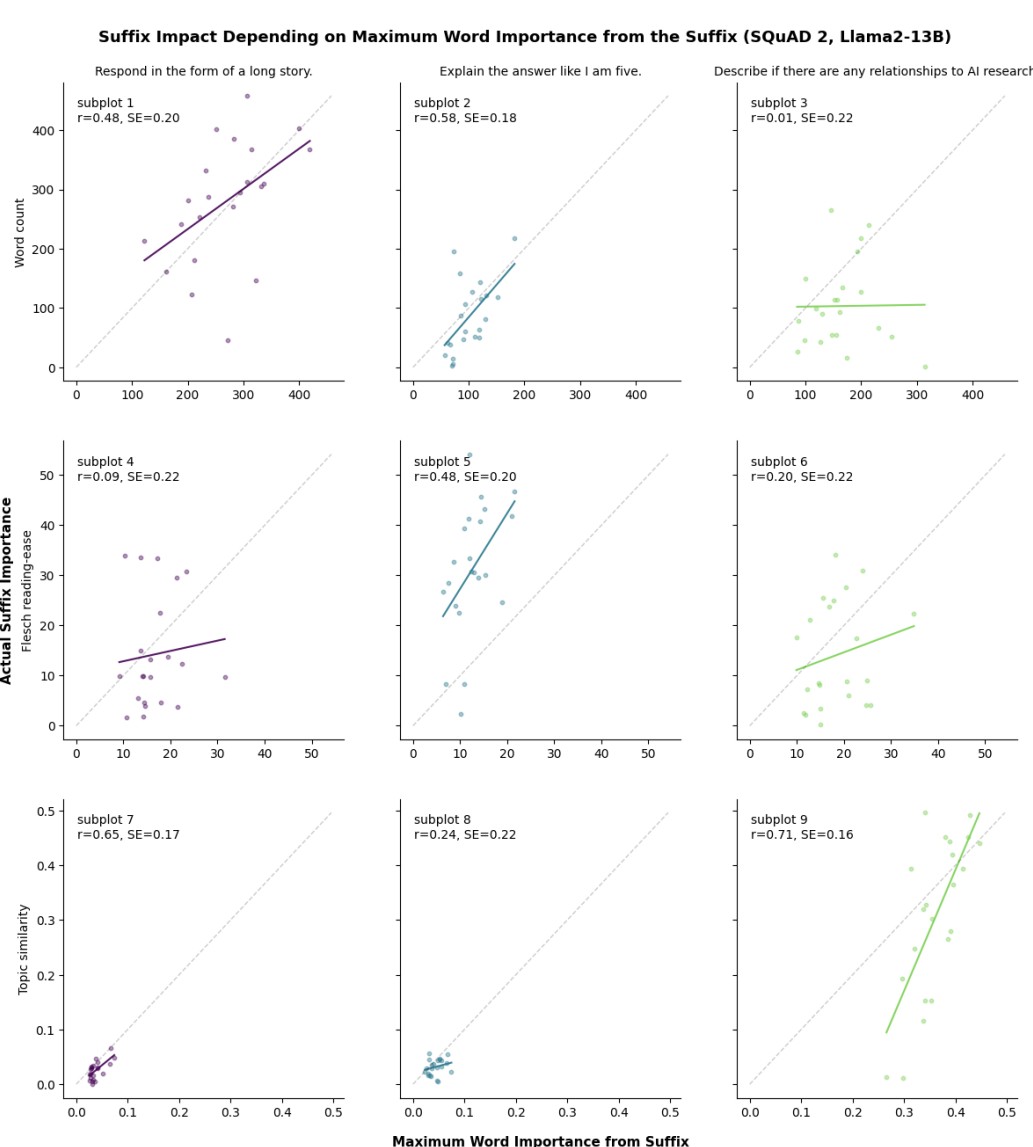

Figure 13: Actual suffix importance vs maximum importance from suffix using Llama2-13B on questions from SQuAD 2. Each column in this plot corresponds to one suffix. For each suffix, the word importance has been calculated using scoring functions "word count", "Flesch reading-ease", and "topic similarity".

