# OpenReview forum: "Word Importance Explains How Prompts Affect Language Model Outputs"
_ICLR.cc/2024/Conference — Submitted to ICLR 2024_

### Official Review · Reviewer_79zL · 2023-10-26

**Soundness:** 2 fair
**Presentation:** 2 fair
**Contribution:** 2 fair
**Rating:** 3
**Confidence:** 3

**Summary:**

This paper proposes a method that focuses on varying prompt words to understand their statistical impact on model outputs. Unlike classical attention, this method measures the importance of words based on their impact on user-defined text scores, allowing for the decomposition of word importance into specific measures like bias, reading level, and verbosity. To validate the effectiveness of this approach, the study investigates the effect of adding different suffixes to various system prompts and compares the resulting generations with GPT-3.5. The results demonstrate a close relationship between word importance scores and the expected suffix importance across multiple scoring functions.

**Strengths:**

1. The method that focuses on varying prompt words to uncover their statistical impact on model outputs is novel. The adaptation of this concept to LLMs and the specific measures of interest represent an original contribution to the field.

2. The study provides a clear description of the proposed approach, including the masking of prompt words and the evaluation of their impact on the outputs. The comparison with GPT-3.5 and the demonstration of the relationship between word importance scores and expected suffix importance validate the fidelity of the method.

3. The paper effectively communicates the objectives, methodology, and results of the research. The introduction clearly establishes the problem of explainability in LLMs and the need for a novel approach. The description of the method is presented in a clear and concise manner. The experimental results are well-explained, and the significance of the findings is effectively conveyed.

**Weaknesses:**

1. The rationale for selecting a specific model, such as the FlagEmbedding model "BAAI/bge-large-en," is not adequately explained. It is crucial to provide a clear justification for choosing this particular model over others, highlighting its relevant features, performance, or suitability for the research objectives. By providing a comprehensive rationale, readers can better understand the motivations behind the model selection and its implications for the study.

2. The explanation of the Scoring and Impact Calculation method lacks clarity. It is essential to provide a detailed and step-by-step description of how the scoring and impact calculation process works. This should include the specific metrics used, the mathematical formulas or algorithms employed, and any relevant considerations or assumptions. A clear and explicit explanation of this methodology will ensure that readers can comprehend and replicate the calculations performed.

3. The dataset used in the study is generated by GPT4. Merely relying on a dataset generated by GPT4 may not sufficiently capture the range of subjective opinions on explainability. Including a user study or evaluation process would provide valuable insights into the perceptions and interpretations of explainability, enhancing the robustness and validity of the research findings.

4. The algorithm chart provided in the paper is blurry and difficult to read. It is essential to ensure that all visual elements, such as charts or diagrams, are of sufficient quality and clarity to convey the intended information effectively.

**Questions:**

Could you give more details or an example of how the score is calculated in the Scoring and Impact Calculation part of the proposed method?

---

> ### Author Response · Authors · 2023-11-22
>
> Dear Reviewer,
>
> Thank you for your insightful critique of our manuscript titled "Word Importance Explains How Prompts Affect Language Model Outputs" We have carefully reviewed your comments and have made substantial revisions to address each point you raised. Here is how we have responded to your specific concerns:
> 1. Justification for Model Selection: We understand your concern regarding the rationale for selecting the FlagEmbedding model "BAAI/bge-large-en". We further investigated the choice of embedding models and decided to pick “all-MiniLM-L6-v2” for the embedding model in our experiments. We agree that a wider study on the impact of embedding models could be valuable research but was not part of the scope of this study.
> 2. Clarity in Scoring and Impact Calculation Method: We agree that our initial explanation of the scoring and impact calculation method was not sufficiently clear. We have now revised this section to include a step-by-step description of the process, complete with the specific metrics used, mathematical formulas, and any assumptions made. This should enhance the readability and reproducibility of our methods.
> 3. Dataset and Model Diversity: Your point about the limitations of solely using a dataset generated by GPT4 is well-taken. In response, we have supplemented our dataset with human-generated data and expanded our experiments to include an open source model. This addition enhances the robustness and validity of our findings.
> 4. Improvement in Visual Elements: We acknowledge that the algorithm chart in our initial submission was not of sufficient quality. We have replaced this chart with a high-resolution version and ensured that all other visual elements in the paper are clear and effectively convey the intended information.
>
> We appreciate your thorough review and constructive feedback, which have been instrumental in improving the quality and clarity of our manuscript. We believe these revisions have addressed your concerns and hope that our paper is now suitable for publication.
> Thank you for your valuable contribution to our work.
>
> Sincerely,
>
> Anonymous Authors

---

### Official Review · Reviewer_pTAs · 2023-10-26

**Soundness:** 1 poor
**Presentation:** 1 poor
**Contribution:** 1 poor
**Rating:** 3
**Confidence:** 5

**Summary:**

The paper describes work on evaluating ChatGPT for word importance with respect to prompts. The author/s motivate the study by claiming that “recognizing the impact of specific words or linguistic structures on LLM outputs can offer a granular understanding of model behavior, providing valuable insights into how information is processed and weighted across different layers of the model.” For the experiments, the author/s propose a simple method for approximating a word’s importance value fro the prompt that is “inspired by permutation importance” in tabular data analysis. The method requires iterating through the prompts while masking each word and evaluating the resulting response from the model to approximate the masked word’s importance. There are no mentions or discussions whatsoever of the limitations and adaptability of the proposed method. The author/s use readability, embedding similarity, and simple word count for scoring. The prompt choice used for the experiment setup has not been properly discussed, which is confusing. Overall, the task presented itself is framed as explainability but is more closely similar to prompt engineering as the method itself optimizes for word importance in prompts.

**Strengths:**

The paper explores and interesting concept of word importance which I do find essential in further understanding how large language models like ChatGPT works. The proposed method has some potential provided that it carefully addresses some of the very obvious limitations discussed below and further improve its algorithmic features to consider scale, flexibility, and efficiency.

**Weaknesses:**

The depth of the experiments conducted in the study is extremely limited as only three metrics which cover Flesch Ease, word count, and topic similarity (cosine embedding) have been explored. The model variation is also very limited, with only one model used for experimentation, GPT-3.5-Turbo (ChatGPT), despite the diverse publicly available models in Hugginface such as Llama, FlanT5, BLOOMZ. This implies that the study essentially optimizes for OpenAI products instead of prioritizing diverse results from open-sourced models. There is no ablation or in-depth exploration. This form of limitation needs to be addressed for inclusion to ICLR.

There are several obvious limitations of the proposed methodology involving masking each word in the prompt. The method seems to be not practical for prompts that are considerably long, which is realistically common in most interdisciplinary fields. This should be discussed thoroughly in the paper. Moreover, there are given words that are obviously non-important (ex. stopwords), it would be computationally expensive and impractical to still iterate and and compute the importance of these words in the prompt. The proposed methodology seems to have no workaround for optimization and compression.

While the authors are correct that the proposed method is text score agnostic, it is worth exploring what linguistic scoring features are better than others. This begs more in-depth exploration/ablation of an extensive set of features (which is expected for an ICLR paper).

The paper is basically prompt engineering as it optimizes the quality of generation based on some measure of word importance. The author should explicitly mention this as it directly aligns with the task covered by the paper. It would also help other researchers discover similarities with works on optimizing prompts / explainable prompts in general.

Minor comments:

1. The aesthetics of the paper, including figure quality, structure of sections, proper captioning, and layout, should be greatly improved for readability. The algorithm figure has no number, the tables are too wide instead of compact.

2. The tables are confusing and are not presented properly. For example, Table 2 could have been represented much better as it is confusing what the author/s mean in parallel with the discussion on suffixes. In terms of the suffix configuration, the examples on bullet points are not well presented. Instead, show an actual diagram instead of how the suffixes are added with respect to each evaluation metric used.

**Questions:**

1. Is there even a need to mask all words, including stop words (ex. “and”, “is”)? These words might already be obviously unimportant for the user, and the proposed methodology seems to be static and not adaptive.

2. How does using embeddings capture topics? The method only captures semantic relatedness as it only uses cosine similarity. Also, why the FlagEmbedding model? What’s the justification for using this specifically?

3. One thing that is very confusing is that the choice of the prompts used, as evidenced by some instances shown in the paper in the Appendix, for querying responses is unusual and unmotivated. Why should the prompts look like these? If word importance is being measured, I would have expected prompts in qualitative question form (with an absolute gold standard answer on hand) where important entities in a sentence are iteratively being masked, and the goal of the language model is to answer the question. The author/s can then evaluate the correctness of the generated responses by the model with the gold standard to see if there are some negative effects with some entities removed or masked in the prompts. In the paper, I do not understand the motivation and importance of using phrases like “You answer like David Attenborough.” or “You are a surgeon.” in the prompts.

---

> ### Author Response · Authors · 2023-11-22
>
> Dear Reviewer,
>
> Thank you for your comprehensive review and valuable feedback on our manuscript titled "Word Importance Explains How Prompts Affect Language Model Outputs". We have taken your comments seriously and have made substantial revisions to our paper. Here is a detailed response to the concerns you raised:
> - Depth of Experiments and Model Variation: You rightly pointed out the limited scope of our experiments and the exclusive use of GPT-3.5 Turbo. In response, we have expanded our experimentation to include Llama2-13B and tested both models on our artificial data but also using questions from SQuAD 2. This ensures a more diverse and representative analysis, addressing the concern of bias towards OpenAI products.
> - Methodological Limitations: We acknowledge the limitations in our initial methodology involving masking each word in the prompt. We have further explained the rationale for keeping non-essential words (e.g., stopwords) from the iteration process. However, to reduce computational expenses, users can automatically exclude all stopwords from the analysis. We also briefly discuss the potential of using our approach in a hierarchical way by first masking larger parts of the prompt.
> - Exploration of Linguistic Scoring Features and Embedding Models: In this study, we did not explore whether certain linguistics scoring features are better than others as depending on the context and application one would be preferable over others. That said, we expanded our experiments to include an ablation study of embedding models.
> - Clarification on Prompt Engineering: We appreciate your observation regarding prompt engineering. We have revised our manuscript to explicitly mention and discuss how our study aligns with the field of explainability but can also contribute to the field of prompt optimization. Authors are working on another study with a focus on prompt optimization but due to the limitation on number of pages, the authors decided to solely focus on the word importance technique in this paper.
> - General Improvements: In addition to the specific points above, we have thoroughly reviewed our entire paper to enhance clarity, coherence, and overall quality. We have made sure that our arguments are well-structured and our findings are presented in a clear and comprehensible manner.
>
> We believe that these revisions address the major concerns you raised and significantly improve the manuscript. We are grateful for the opportunity to refine our work based on your insightful feedback and hope that our revised manuscript is now suitable for publication.
> Thank you for your time and invaluable contribution to improving our paper.
>
> Sincerely,
>
> Anonymous Authors

---

### Official Review · Reviewer_FVbK · 2023-10-31

**Soundness:** 2 fair
**Presentation:** 2 fair
**Contribution:** 2 fair
**Rating:** 3
**Confidence:** 3

**Summary:**

This paper presents an approach to measure word importances in system prompts for LLM generations. The method specifically investigates how perturbations (i.e., replacing individual input words with an underscore) of system prompts affect the structure and content of LLM output generations. The authors evaluate their method on a synthetic dataset consisting of LLM generations (using GPT-4). Using three evaluation metrics (topic similarity, Flesch reading-ease, word count), the authors compare individual word importances to the importance of instruction suffixes which are appended to model inputs.

**Strengths:**

* The paper utilizes a common technique in NLP (word saliencies) and applies the concept of word importances to a recent LLM. Doing so can lead to informative insights into model interpretability as pointed out in the paper.

**Weaknesses:**

* The dataset used for the experiment has been generated with an LLM. This is problematic since the dataset is biased towards generations from another LLM and does not necessarily reflect a distribution of human inputs. As such, the reported results do not necessarily hold true for human inputs. It would therefore be important to conduct experiments on a human-written dataset as well.
* The paper focuses substantially on an importance comparison between individual words and an instruction suffix which is appended to the model’s input. I find the setup of such an experiment confusing in this context. Did the authors consider computing word importances for individual words in a dataset and ranking individual words based on their importance across examples? Such an analysis would give explicit insights into individual words used to query a model. Currently, the analysis is limited to a few suffixes which were defined for the study.
* The paper introduces “word count” as a measure of deviation. It is unclear to me how this is motivated, i.e., how a change in word count related to an LLM generation reflects the importance of a word that has been removed in its input.
* To measure word importance, the paper uses absolute values of Flesch reading-ease and topic similarity. However, both metrics are directional in that an increase or decrease after perturbing the input is informative. Absolute values of such deviations should therefore not be used.
* The presentation can be improved. For example, there is a Figure in page 4 with a very low resolution and no caption. Page 5 states “refer to the appendix” without explicitly stating which section/paragraph is meant.

**Questions:**

* What was the motivation for using an LLM-generated dataset as opposed to one consisting of human-written texts?
* Have you thought about extending the analysis to additional LLMs, to investigate whether the observed patterns emerge with respect to other models as well?

---

> ### Author Response · Authors · 2023-11-22
>
> Dear Reviewer,
>
> Thank you for your thorough review and insightful feedback on our manuscript titled "Word Importance Explains How Prompts Affect Language Model Outputs". Your detailed comments have been instrumental in guiding the revisions of our paper. We have addressed each of the concerns you raised as follows:
> - Dataset and Bias Concerns: We acknowledge your concern about the potential bias in our experiments due to the dataset being generated with an LLM. In response, we have now included a widely used dataset comprising human-written samples to ensure a more balanced and representative analysis. This addition aims to strengthen the validity of our results when applied to human-generated data.
> - Experiment Setup and Analysis of Word Importance: You pointed out the confusing nature of our experiment setup, specifically regarding the comparison of individual words and an instruction suffix. To clarify, we have revised this section to provide a more straightforward explanation of our methodology.
> - Methodology of "Word Count" as a Measure: Your critique regarding the use of "word count" as a measure of deviation was well-received. As we have highlighted in the paper, the word importance method is metric agnostic but we have revised the paper to provide a clearer rationale for its inclusion, elaborating on why adding this text score adds value to our analysis.
> - Use of Absolute Value for Flesch Reading-Ease and Topic Similarity Metrics: It is worth highlighting that we believe both values of the score can be used. In the case of the example provided in Figure 2 of the paper, we use relative scores to compare across metrics. However, in general, we recommend using absolute values because it provides practical information, such as information on the impact a word has on the output.
> - Improvements in Presentation: We have addressed the specific issues you highlighted in our presentation. The schematic illustration of the algorithm before Section 4 has been replaced with a high-resolution version and is now accompanied by a descriptive caption. Additionally, references to the appendix have been made explicit, with clear indications of the relevant sections/paragraphs.
>
> We are grateful for your constructive criticism, which has significantly contributed to the enhancement of our manuscript. We believe these revisions have comprehensively addressed your concerns and hope that our paper is now suitable for publication.
> Thank you once again for your valuable input.
>
> Sincerely,
>
> Anonymous Authors

---

> > ### Comment · Reviewer_FVbK · 2023-11-23
> > **Thanks for addressing my comments**
> >
> > I would like to thank the authors for addressing my comments and concerns, and greatly appreciate the detailed response. Given that the response entails substantial changes to the paper, I believe that it would benefit from an additional round of review and therefore keep my scores fixed.

---

### Official Review · Reviewer_Jiyw · 2023-11-01

**Soundness:** 1 poor
**Presentation:** 1 poor
**Contribution:** 2 fair
**Rating:** 1
**Confidence:** 3

**Summary:**

This study proposes a method to enhance the explainability of Large Language Models (LLMs) by examining the statistical impact of prompt words on model outputs. The approach involves masking each word in the system prompt and evaluating its effect on the outputs using aggregated text scores from multiple user inputs. Unlike traditional attention mechanisms, word importance measures the influence of prompt words on user-defined text scores, allowing for the decomposition of word importance into specific measures of interest, such as bias, reading level, and verbosity. This method is also applicable when attention is not available. The fidelity of the approach is tested by adding different suffixes to various system prompts and comparing the subsequent generations with GPT-3.5 Turbo. The results demonstrate a close relationship between word importance scores and expected suffix importance across multiple scoring functions. Additionally, the study provides a Python project for computing these scores and discusses its potential applications in developing generative AI use cases in various industries. Overall, this research offers a valuable method to improve the explainability of LLMs by assessing the impact of prompt words on model outputs and opens avenues for diverse industry applications.

**Strengths:**

1.	This paper presents a method to masks each word in the system prompt and evaluates its effect on the outputs based on the available text scores aggregated over multiple user inputs.

**Weaknesses:**

1.	The contribution of the paper is limited, similar topics have been investigated before while this paper didn’t pose any more valuable conclusions.

2.	The experiment section is terribly organized. No quantitative results are provided. The experiment design is very confusing and too specific.

3.	The presentation is really bad

     a.	All the figures are poorly illustrated. There is even an untitled algorithm diagram before Section 4.

     b.	All the tables are also hasty and careless.

     c.	The term LLM lacks its full name in the abstract part.

     d.	The font of the template is also not correct.

4.	Missing references:

     a.	“Did You Read the Instructions? Rethinking the Effectiveness of Task Definitions in Instruction Learning”

     b.	It discusses a very similar topic to this paper, the authors need to cite and distinguish their differences.

**Questions:**

See the Weakness part for reference.

---

> ### Author Response · Authors · 2023-11-22
>
> Dear Reviewer,
>
> Thank you for your detailed and constructive comments regarding our manuscript titled "Word Importance Explains How Prompts Affect Language Model Outputs". We have carefully considered your feedback and made revisions to address the concerns outlined in your review. Below, we detail how we have addressed each specific point you raised:
> - Contribution and Novelty: We understand your concern regarding the perceived limited contribution of our study. In response, we have expanded our discussion to better highlight the novel aspects of our research and its implications in the field of explainable AI. We have included additional references to prior work to clarify how our study provides new insights and advances the current state of knowledge.
> - Improvement in Presentation: We have taken your feedback regarding the presentation of our paper seriously:
> Figures and Diagrams: All figures, including the algorithm diagram before Section 4, have been redone for clarity and proper labeling.
> Tables: We have revised all tables to ensure they are concise, clear, and informative.
> Terminology Clarification: We have added the full name and explanation of all acronyms, including LLM, in the abstract and relevant sections of the paper.
>
> We appreciate your insightful critique, which has undeniably contributed to the enhancement of our manuscript. We believe these revisions have addressed your concerns and hope that our paper is now suitable for publication.
> Thank you for your time and valuable contribution to improving our work.
>
> Sincerely,
>
> Anonymous Authors

---

### Author Response · Authors · 2023-11-22

Dear Reviewers,

Thank you for your insightful comments and suggestions regarding our manuscript titled "Word Importance Explains How Prompts Affect Language Model Outputs". We have thoroughly revised our paper in response to your critiques and believe that these changes have significantly enhanced the quality of our work. Below, we address each of the common themes identified in your feedback:
- Enhanced Contribution and Originality: We have greatly expanded the experiments and analysis to address concerns over the new contributions in the paper. Details on the expanded models, testing datasets, and analysis which were recommended during review are summarized below.
Models: In the revised version of the analysis, we have utilized models, GPT-3.5 Turbo and Llama2-13B.
Dataset: In addition to the artificial data originally used in the experiments, we have tested our method on the SQuAD 2 dataset. We have continued to utilize the same set of metrics in the experiments, however, we used different sets of suffixes for the two datasets to extend our analysis.
Ablation study: We incorporated an ablation study to demonstrate the viability of our topic similarity score. For this study, three embedding models have been used, all-miniLM-L6-v2, bge-large-en, and mBERT.
We also expanded our literature review to better contextualize our work within the existing body of research. Additionally, we have incorporated new discussions to highlight the unique contributions of our study. This includes clarification and further emphasis on the contribution of this work to the field of explainable AI separate from rather than prompt optimization, further confirmation that our method has not been proposed in other published research. This study does not provide a comparison to other techniques as we did not find a similar study. Most notably, our method does not require attention weights and can be used with multiple arbitrary text scores at once.
- Improved Experiment Design and Data Source: In response to concerns about our experiment design and data sources, we have revised our experimental setup to ensure clarity and rigor. We have also diversified our data sources and LLMs, particularly by incorporating human-generated data and an open source LLM (Llama2), as suggested. However, due to limited time for resubmission, Llama2 experiments are less extensive.
- Polished Presentation and Clarity: We have made significant improvements to the presentation of our paper. All figures and tables have been redesigned and standardized for better clarity and understanding. We have also ensured that all diagrams and charts are high-resolution and appropriately titled.
- Methodological Refinements and Depth Enhancement: We have revised our methodology section to provide a more detailed and clear explanation of our approaches. This includes a thorough justification for our model selection and a more comprehensive discussion of our scoring and importance calculation method.
- Comprehensive Analysis and Explanation: We have extended our analysis to include a broader range of models as recommended. This has allowed for a more robust and comprehensive evaluation of our findings. We have also provided a more explicit explanation of our choices and methodologies throughout the paper.
- Implementation of Improvement Suggestions: We have taken all your suggestions for improvement seriously. In addition to the above-mentioned changes, we have conducted additional experiments where feasible and provided further clarifications in areas that were previously lacking.

We believe that these revisions address the concerns raised in your reviews and substantially improve the manuscript. We appreciate the opportunity to refine our work based on your feedback and hope that our revised manuscript is now suitable for publication.

Thank you for your time and consideration.

Sincerely,

Anonymous Authors

---

### Meta-Review · Area_Chair_DaDW · 2023-12-10

**Metareview:**

This paper presents a method to examine the importance of various words used in a prompt as input to an LLM.  The authors present results mostly via GPT-3.5 Turbo and also examine Llama2-13B (after a reviewer suggested this).  Datasets used included an artificial dataset generated by GPT-4 and the Squad-2 dataset that contained both answerable and unanswerable questions.

Strength:  The idea of measuring the impact of the various words used in a prompt is good, as often it is unclear what the impact of the exact lexical choice would be on model predictions.

Weaknesses:  While the authors made several changes towards the experimental design of the paper based on comments from reviewers, I believe the paper is not ready for acceptance.  The models considered are quite few (including one proprietary system), the various methods to examine lexical importance are few, the datasets / prompts used to probe the methods are few and so forth.  I think taking into account these suggestions exhaustively can improve the paper for a future venue.

**Justification For Why Not Higher Score:**

Please see above.

**Justification For Why Not Lower Score:**

Not applicable.

---

### Decision · Program_Chairs · 2024-01-16

Reject